# YOLO-DRS: A Bioinspired Object Detection Algorithm for Remote Sensing Images Incorporating a Multi-Scale Efficient Lightweight Attention Mechanism

**DOI:** 10.3390/biomimetics8060458

**Published:** 2023-10-01

**Authors:** Huan Liao, Wenqiu Zhu

**Affiliations:** School of Computer Science, Hunan University of Technology, Zhuzhou 412007, China; m21077500009@stu.hut.edu.cn

**Keywords:** bioinspired object detection, YOLOv5, multi-scale, attention mechanisms, transposed convolution

## Abstract

Bioinspired object detection in remotely sensed images plays an important role in a variety of fields. Due to the small size of the target, complex background information, and multi-scale remote sensing images, the generalized YOLOv5 detection framework is unable to obtain good detection results. In order to deal with this issue, we proposed YOLO-DRS, a bioinspired object detection algorithm for remote sensing images incorporating a multi-scale efficient lightweight attention mechanism. First, we proposed LEC, a lightweight multi-scale module for efficient attention mechanisms. The fusion of multi-scale feature information allows the LEC module to completely improve the model’s ability to extract multi-scale targets and recognize more targets. Then, we propose a transposed convolutional upsampling alternative to the original nearest-neighbor interpolation algorithm. Transposed convolutional upsampling has the potential to greatly reduce the loss of feature information by learning the feature information dynamically, thereby reducing problems such as missed detections and false detections of small targets by the model. Our proposed YOLO-DRS algorithm exhibits significant improvements over the original YOLOv5s. Specifically, it achieves a 2.3% increase in precision (P), a 3.2% increase in recall (R), and a 2.5% increase in mAP@0.5. Notably, the introduction of the LEC module and transposed convolutional results in a respective improvement of 2.2% and 2.1% in mAP@0.5. In addition, YOLO-DRS only increased the GFLOPs by 0.2. In comparison to the state-of-the-art algorithms, namely YOLOv8s and YOLOv7-tiny, YOLO-DRS demonstrates significant improvements in the mAP@0.5 metrics, with enhancements ranging from 1.8% to 7.3%. It is fully proved that our YOLO-DRS can reduce the missed and false detection problems of remote sensing target detection.

## 1. Introduction

With the rapid development of bioinspired image processing and remote sensing technology, remote sensing object detection technology has gradually become a hot spot in current research. It is widely used in the fields of national defense, rescue [1], urban construction, geologic disasters [2], and development. In remote sensing imagery, the task of target detection is to detect and identify the precise location of specific categories of targets, such as common aircraft, automobiles, oiltank, playgrounds, etc., in remotely sensed imagery. For remote sensing images, the targets in these images are usually densely distributed, have too many small-sized targets distributed at multiple scales, and can be affected by factors such as the complexity of the detection background. Initially, features were typically extracted by artificial means such as classical algorithms, such as AdaBoost [3], SVM [4], HoGDetector [5], DMP [6], etc. However, these algorithms perform poorly in complex settings, and the high algorithm complexity makes detection inefficient and time-consuming.

The convolutional neural network(CNN) based on deep learning [7] performed well in the ImageNet image classification competition in 2012, which led to the rapid development of convolutional neural networks. In target detection, the convolutional neural network is the main direction of target detection. At present, there are two kinds of target detection methods based on deep learning. One class is two-stage target detection methods based on candidate frames, such as R-CNN [8], Fast R-CNN [9], Faster R-CNN [10], and Mask R-CNN [11] algorithms, which are more complex in design, consume more resources, and have slower detection speeds and do not meet the requirements of real-time detection. Another is a single-level regression-based target detection algorithms representing the SSD [12], Retina-Net [13], CenterNet [14], and YOLO [15,16,17,18] series. Compared with the two-stage target detection algorithm, the single-stage target detection algorithm not only has high detection accuracy but also greatly improves the detection speed, so it has been more widely used.

In recent years, the YOLO series has become a representative algorithm in the field of target detection with its fast, accurate, and mature engineering capabilities. The YOLO family of algorithms has been repeatedly improved and optimized, and while it has now evolved into YOLOv8, the YOLOv5 algorithm is the most widely used and mature algorithm for both academic and industrial use.

## 2. Related Work

In recent years, numerous scholars have made significant advancements in deep-learning-based object detection methods. Farhan Ullah et al. [19] proposed a cyber threat detection system that combines migration learning and multi-model image characterization in a hybrid approach. Du et al. [20] introduced BV-YOLOv5S, a modification of YOLOv5S, to achieve real-time defect detection in road pits. Li et al. [21] developed a lightweight convolutional neural network called WearNet. This network is designed to enable real-time detection of scratches on sliding metal parts. Shen et al. [22] focused on enhancing cross-scale detection in road object detection tasks using the YOLOv3 model. They employed the K-means-GIoU algorithm to generate prior boxes and implement a detection branch specifically for small targets. Wang Jian et al. [23] analyzed the challenges posed by high resolution and complex backgrounds in UAV aerial images. To address these issues, they proposed MFP-YOLO, a lightweight detection algorithm based on YOLOv5. This method combines a multi-path inverse residual module and attention module. Additionally, it utilizes a parallel deconvolutional space pyramid pool to extract scale-specific information, thereby improving the detection performance of the algorithm. Furthermore, many scholars have made significant breakthroughs in the field of remote sensing image object detection. Using the YOLOv3 model, Qu et al. [24]. proposed an auxiliary network to improve the recognition of objects in remote sensing images. The CBAM module is backward-compatible to improve network performance and prevent the loss of crucial information during training. Reference [25] proposed using DenseNet [26] to enhance YOLOv3 and to improve the accuracy of remote sensing image detection by enhancing the structure in the backbone. However, DenseNet’s structure is too complex and has too many parameters, leading to a drop in detection speed. Reference [27] introduced lightweight enhancements to the structure of YOLOv3 as well as an introduction to Res2Net [28] to improve the accuracy and speed of remote sensing target detection. In reference [29], the PPM (pyramid pooling module) [30] was added based on YOLOv4, and the Mish function was used to override the original activation function, which improved the detection precision and recall rate of aircraft and dockyards in remotely sensed imagery. Li et al. [31] proposed the YOLOSR-IST model. Based on the YOLOv5 method, this model introduces coordinate attention during the feature fusion process and integrates high-resolution maps.

However, these methods above do not give reasonable solutions for the problems of false detection and missed detection that occur in remote sensing image target detection. To address these challenges, we designed a remote sensing target detection algorithm YOLO-DRS based on YOLOv5. Our work makes the following main contributions.

Based on the original EMA attention, a new module LEC(LDW-EMA-C3) is proposed for the fusion of a multi-scale lightweight efficient attention with the C3 structure in YOLOv5, replacing the last two C3 modules of the backbone with LDW-EMA to extract high-dimensional feature information at different scales.In the upsampling process of YOLOv5, the upsampling transposed convolution is introduced to replace the original nearest-neighbor interpolation upsampling to reduce the loss of the feature information of small targets in the upsampling process.

The rest of the paper is organized as follows: Section 3 introduces the YOLOv5 method. Section 4 introduces the methods proposed. Section 5 introduces the experimental part. Section 6 concludes with a summary of the paper.

## 3. The Basic Structure Of YOLOv5s

YOLOv5 is available in four different sizes based on depth and width: YOLOv5s, YOLOv5m, YOLOV5l, and YOLOv5x. As the model depth deepens and the width increases, YOLOv5 improves its detection accuracy, but the speed of detection decreases along with it. In this paper, we have selected YOLOv5s version 6.1, which combines both detection speed and accuracy advantages. YOLOv5s is mainly composed of four parts: input module (Input), backbone network module (Backbone), feature fusion module (Neck), and prediction module (Head). The overall architecture of YOLOv5s is shown in Figure 1.

The input side works as follows: first, a group of up to four images is scaled, aligned, or cropped to form a single image after capturing the enhanced image mosaic data. Secondly, the YOLOv5 algorithm adjusts the black edge by equidistant scaling and filling the smallest black edge with the smallest black edge, thus unifying the size of the image and preparing the neural network model for training. Figure 2 shows the picture enhanced by Mosaic4 data at the inputs.

The YOLOv5 backbone network consists mainly of the CSP, CBS, and SPPF structures. The CSP structure mainly draws on the idea of the cross-stage network CSPNet [32], where the input features are processed in two parts. The main part extracts features step by step through convolution, normalization, and activation functions, and the branches simply adjust the channels through convolutional layers. By dividing the gradient information, a large amount of redundant gradient information is eliminated. The CBS structure consists of a convolution, Conv, a normalized BatchNorm [33], and an activation function, SiLU [34], which is used to extract the features of the model. The SPPF structure serially passes the input features through multiple 5 × 5 maximal pooling layers and then extracts the stacked features via the CBS network structure, which can increase the receptive field of the network and enhance the network’s characterization capability.

The feature fusion module (Neck) is mainly composed of feature pyramid network (FPN) [35] and path aggregation network (PAN) [36] modules, which are responsible for fusing the feature maps of various scales and then decoding and generating feature maps containing more semantic information for input to the prediction module.

The YOLOv5s prediction module consists of three detection layers of different scales, 80 × 80, 40 × 40, and 20 × 20, which are used to predict the category and position prediction of small, medium, and large targets. The category information of the objects with the highest confidence scores is then output through post-processing operations, such as the non-maximum suppression algorithm.

YOLOv5s loss functions include cls_loss, box_loss, and obj_loss. The cls_loss and obj_loss are calculated using BCEWithLogitsLoss as shown in Equation (Equation 1).
(1)$$C=-\frac{1}{n}\sum_{x}\left[ylna+\left(1-y\right)ln\left(1-a\right)\right]$$
where *x* denotes the sample, y denotes the label, a denotes the predicted output, and *n* denotes the total number of samples.

The box_loss is calculated via the IoU function [37]. The schematic and equations are shown in Figure 3 and Equation (Equation 2). YOLOv5s version 6.1 uses CIoU loss, as shown in Equation (Equation 3).
(2)$$IOU=\frac{A\cap B}{A\cup B}$$
where A∩B is the area of overlap between the real frame and the predicted frame, and A∪B is the total area between the two.
(3)$$CIoU=1-\frac{\rho^{2}\left(A,B\right)}{c^{2}}+\alpha v$$
where $\rho^{2}\left(A,B\right)$ represents the Euclidean distance between the centers of the predicted frame *A* and the real frame *B*, *c* represents the diagonal length of the smallest rectangle containing *A* and *B*, α represents the weight parameter, and *v* is used as a measure of the variability of the length, width, and height.

## 4. Proposed Method

The overall architecture of our proposed YOLO-GCRS is shown in Figure 4. First, the multi-scale feature information of the image can be fully extracted by the LEC module in the backbone network part. Then, the loss of small target feature information is reduced by the transposed convolutional upsampling method in the Neck section. Finally, the output target and category information is carried out through the prediction header.

### 4.1. The LEC Module

With the development of deep convolutional neural networks, the attention mechanism has attracted great interest from the computer vision research community. The flexible structural features of the attentional mechanism approach not only enhance the learning of more discriminative feature representations but can also be easily inserted into the backbone architecture of neural networks.

It is widely recognized that there are three main mechanisms of attention that have been proposed, such as channeled attention, spatial attention, and both. As a representative of channel attention, SE [38] explicitly models cross-dimensional interactions to extract channel attention. The convolutional block attention module (CBAM) [39] builds cross-channel and cross-spatial information with semantic interdependencies between spatial and channel dimensions in the feature map. However, modeling cross-channel relationships using channel dimensionality reduction may introduce side effects when extracting deep visual representations. To solve these problems, Daliang Ouyang et al. proposed a new Efficient multiscale attention (EMA) [40] module by modifying the sequential processing of the CA [41] attention mechanism.

The general structure of EMA is shown in Figure 5. On the one hand, two coded features are connected in the image height direction and made to share the same 1 × 1 convolution without dimensionality reduction in the 1 × 1 branch by a similar process as CA. After decomposing the output of the 1 × 1 convolutional into two vectors, two nonlinear Sigmoid functions are used to fit a 2D binary distribution on the linear convolution. To realize different cross-channel interaction features between two parallel routes in a 1 × 1 branch, the two-channel attention maps within each group are aggregated together by simple multiplication. On the other hand, the 3 × 3 branch captures local cross-channel interactions via 3 × 3 convolutional to expand the feature space. In this way, EMA not only encodes inter-channel information to adjust the importance of different channels but also saves precise spatial structure information into the channels.

Furthermore, a cross-spatial learning strategy is proposed in the EMA article, which is designed to encode global information and model long-range dependencies. For efficient computation, the natural nonlinear function Softmax for 2D Gaussian mapping is used at the output of the 2D global mean pooling (Avg Pool) to fit the linear transformation. The first spatial attention map was derived by multiplying the output of the above parallel processing with the matrix dot product operation. Similarly, 2D global average pooling is utilized to encode the global spatial information in the branch to derive a second spatial attention map that preserves the entire precise spatial location information. Finally, the output feature maps within each group were computed as an aggregation of the two generated spatial attention weight values. The sigmoid function captures pairwise relationships at the pixel level and highlights the global context of all pixels. The final output of the EMA is the same size as the input, which is efficient for stacking into modern architectures. The 2D global pooling and Softmax function formulas are (Equation 4) and (Equation 5), respectively.
(4)$$Z_{C}=\frac{1}{H\times W}\sum_{j}^{H}\sum_{i}^{W}X_{C}\left(i,j\right)$$
where H, W, and C represent the height, width, and dimension of the input feature map, respectively.
(5)$$Softmax\left(z_{i}\right)=\frac{e^{z_{i}}}{\sum_{c=1}^{C}e^{z_{c}}}$$
where $z_{i}$ is the output value of the ith node, C is the number of output nodes, and e is a constant term.

In order to extract the multi-scale feature information in the complex background without increasing the computation cost too much, first, in this paper, we propose a lightweight convolutional block, LDW. Then, based on LDW, we propose a lightweight multi-scale efficient attention mechanism module, LDW-EMA. Figure 6 and Figure 7 show the detailed structures of LDW and LDW-EMA, respectively.

The LDW convolutional block consists of Conv 1 × 1, DW3 × 3, BatchNorm, and ReLU activation functions. DSC (Deep separable convolution) consists of Conv 1 × 1 and DW 3 × 3. DSC dramatically reduces the convolutional parameters. BatchNorm speeds model convergence and improves the stability of the model. The ReLU activation function increases network non-linearity and prevents the gradient from disappearing. In addition, we use two DW 3 × 3 instead of one DW 5 × 5. This is because two DW 3 × 3 can achieve the same effect as one DW 5 × 5 with smaller parameters.

The LDW-EMA is composed of four branches: the principal branch, the coordinate branch, the 3 × 3 LDW, and the 5 × 5 LDW. Firstly, we used LDW 3 × 3 to replace the initial Conv 3 × 3. Then, we added a new 5 × 5 LDW branch and merged features from that branch with coordinate branch features for learning purposes. The new-cross-spatial learning consists of coordinate branches, LDW 3 × 3 and LDW 5 × 5, which can efficiently learn more multi-scale feature data.

Next, we fuse the proposed LDW-EMA with C3 of the YOLOv5 model Backbone to form the new module LEC. The overall structure of the LEC is shown in Figure 8. We replace the two C3 structures behind the Backbone layer with LEC, this is because the deeper features of the model are more difficult to extract and require more attention mechanisms to help, the shallow features can generally be extracted by the model more easily and accurately.

### 4.2. Transposed Convolution

The original upsampling process of YOLOv5 used the up-adoption method of nearest-neighbor interpolation. In this method, upsampling is used where neighboring pixels are filled with blanks. In high-altitude remote sensing images, because background information is too complex and small targets occupy too many pixel points, the uppermost method of nearest-neighbor interpolation is equivalent to adding too much complex background information.

Transposed convolution can dynamically learn network-based weighting parameters, instead of fixing the use of a particular interpolation method when performing upsampling. Back in semantic segmentation, features would be extracted with a convolutional layer in the encoder, and then the original dimensions would be recovered in the decoder to categorize each pixel in the original image, a process that also requires transposed convolution. The classical methods are FCN [42] and U-Net [43].

The operation steps of transposed convolution can be divided into the following.

Fill rows s-1 and column 0 between input feature mapping elements (where s denotes stride to transform convolution).Fill k-p-1 rows and column 0 around the input feature map (where k denotes the kernel_size size of the transposed convolutional and p is the padding of the transposed convolution).Flip the convolutional kernel parameters up and down, left and right.Perform normal convolution operations (padding = 0, stride = 1).

The following assumes that the input feature map is of size 2 × 2 (assuming that the input and output are single channels), and a feature map of size 4 × 4 is obtained after convolution by transposition(kernel_size = 3, stride = 1, padding = 0, ignore bias). Figure 9 shows the detailed execution of transposed convolution.

First, fill s-1 = 0 rows and column 0 (equal to 0 without padding) between elements.Second, fill k-p-1 = 2 rows and columns around the feature map 0.Third, the convolutional kernel parameters are flipped up and down, left and right.Finally, perform normal convolutional (padding = 0, stride = 1).

The size of the feature map after the transposed convolution operation can be calculated by Equations (Equation 6) and (Equation 7).
(6)$$H_{out}=\left(H_{in}-1\right)\times stride\left[0\right]-2\times padding\left[0\right]+kernel\_size\left[0\right]$$
(7)$$W_{out}=\left(W_{in}-1\right)\times stride\left[1\right]-2\times padding\left[1\right]+kernel\_size\left[1\right]$$
where stride[0] denotes stride in the height direction, padding[0] denotes padding in the height direction, kernel_size[0] denotes kernel_size in the height direction, and index [1] indicates width direction.

In this paper, we introduce the upsampling method of transposed convolution replacing the original nearest-neighbor interpolation. The transposed convolution can reduce the information loss when sampling small targets in the feature map as a way to solve problems such as missed detection and false detection of small targets in remote sensing images.

## 5. Experiments

In this paper, the experimental environment is shown in Table 1.

### 5.1. Datasets

RSOD is the remote sensing dataset employed in the experiment. RSOD is a publically available target detection dataset released by Wuhan University. There are four categories in the dataset: aircraft, playground, overpass, and oiltank. RSOD is labeled according to the PASCAL VOC dataset format.

Table 2 shows in detail the type and number of datasets.

In addition, the sample RSOD dataset and dataset characteristics are shown in Figure 10 and Figure 11. As can be seen from Figure 10 and Figure 11, the sample dataset has too many small target sizes and complex background information and is characterized by multi-scale distribution.

### 5.2. Evaluation Metrics

#### 5.2.1. Precision

Precision is the rate of correct predictions among all results predicted for positive samples.
(8)$$Precision=\frac{TP}{TP+FP}$$
where true positive (TP) means that the prediction is a positive example and the label value is also a positive example, and false positive (FP) means that the prediction is a positive example and the label value is a negative example.

#### 5.2.2. Recall

Recall denotes the probability that of all the outcomes predicted to be positive samples, it is really a positive sample.
(9)$$Recall=\frac{TP}{TP+FN}$$
where false negative (FN) indicates that the prediction is a negative example and the labeled value is a positive example.

#### 5.2.3. Mean Average Precision

The mAP represents the average precision (AP) averaged over all categories.
(10)$$mAP=\frac{1}{N}\sum AP_{i}$$
where *N* represents the total number of categories and $AP_{i}$ represents the average precision in category *i*.

mAP@0.5 denotes the average accuracy value of the IoU parameter when selected as a 0.5 threshold.

#### 5.2.4. FLOPs

FLOPs (floating-point of operations) is the number of floating-point operations, understood as the amount of computation, which can be used to measure algorithm complexity.

#### 5.2.5. FPS

FPS is defined in the field of graphics as the number of frames transmitted per second of the picture. The FPS unit is frame/s.
(11)$$FPS=\frac{Frames}{Time}$$

In this experiment, the FPS on the GPU was selected as the criterion.

### 5.3. Network Training and Parameter Setting

#### 5.3.1. Parameter Setting

In this paper, the detailed training parameter settings are shown in Table 3.

Where yolov5s.pt comes from the pre-training weights learned from ImageNet migration, and the division ratio is the proportion of dataset division.

#### 5.3.2. Network Training

The loss function curve shows the results of network training in the most straightforward way. In this paper, the loss function consists of three main components: cls_loss, box_loss, and obj_loss.
(12)$$L_{loss}=L_{cls}+L_{obj}+L_{box}$$
where $L_{cls}$,$L_{obj}$ and $L_{box}$ represent cls_loss, obj_loss, and box_loss, respectively.

Therefore, we can tell how well the network is trained by observing these three types of loss function images. The loss function curves for each category are shown in Figure 12. From the visualization results, it can be concluded that the YOLO-GCRS model loss decreases with the increase in the number of iterations, and the loss value tends to be stable and close to 0 after the number of iterations reaches 80, indicating that the model training has reached the optimal effect.

### 5.4. Analysis of Results

In this paper, we have conducted extensive ablation experiments to demonstrate the effectiveness and sophistication of the designed module. So, as you know, the ablation experimental data were obtained on the validation set.

Firstly, we discuss the embedding location of the LDW-EMA module in the LEC structure. We name the cases where LDW-EMA is added to the residual structure branch of the LEC structure, the CBS branch, and both as LEC-top, LEC-bottom, and LEC-both, respectively. Table 4 shows detailed experimental data on the different positions of LDW-EMA in the LEC structure.

Overall, it is clear that the LEC-top is an optimal outcome across all metrics. In particular, LEC-top achieved the best results on mAP@0.5. Therefore, we finally chose LEC-top as the structure of the LEC module.

Secondly, the activation function of LDW is discussed after determining the LEC locational structure. This is because the activation function has an important influence on the convergence and training effect of the model. We discuss the activation functions of RELU, Mish, and SILU. Table 5 shows the results of detailed experimental data on different activation functions in the LDW structure.

It can be seen that the model is able to obtain the highest mAP@0.5 when the LDW module uses the ReLU activation function. Therefore, we choose ReLU as the activation function of LDW.

Thirdly, the position of the sample on the transformation convolution is discussed. The nearest-neighbor upsampling interpolation algorithm is replaced by a bottom-up transposed convolution, including replacing the former, replacing the latter, and replacing all. We named them Trans-first, Trans-second, and Trans-both. Table 6 shows the experimental data of transposed convolution at different locations.

For this experiment, we selected Trans-both as the transposed convolution structure. This is the result of synthesizing Trans-both in precision, recall, and mAP@0.5. Although the Trans-2 mAP@0.5 is not the best.

Next, we discuss various cases of EMA in relation to each other. Among them, LDW (3 × 3) denotes the replacement of the convolutional of 3 × 3 branches in EMA with our proposed LDW module. Table 7 shows the detailed data of the experiments for different cases of EMA. It should be noted that these data all have the same structure as the LEC except that the mechanisms of integrated attention are different.

Clearly, our proposed LDW-EMA performs well on precision, recall, and mAP0.5. And the LDW-EMA achieves the highest value of mAP.

We then compare the LEC module proposed in this paper with mainstream attention mechanisms, such as CA, SE, and ECA [44]. We use the YOLOv5s base model in conjunction with each attention mechanism separately. Notably, to ensure the same structure as the LEC, we also integrated the various attentional mechanisms with the two C3 structures at the back of the YOLOv5s backbone. Table 8 shows detailed experimental data for the various different attentional mechanisms.

It is easy to find that LEC achieves the optimal result on the mAP0.5 metric, and collectively, P and R also perform very well, which is sufficient to show that the LEC module we designed embodies better than other mainstream attention mechanisms.

In addition, in order to visualize the practicality of the innovations in each module of this paper, we conducted ablation experiments on the YOLO-DRS algorithm on the validation set. Table 9 demonstrates the detailed data of the ablation experiments.

Analyzing the data in Table 9, the LEC and transposed convolution proposed show a large improvement over the original YOLOv5s in P, R, and mAP@0.5.

Moreover, our proposed YOLO-DRS algorithm improves **2.3%**, **3.2%**, and **2.5%** on P, R, and mAP@0.5, respectively, compared with the original YOLOv5s, and the GFLOPS increases only by 0.2. Also, the FPS is within the real-time detection frame rate range. These data fully prove that the proposed YOLO-DRS algorithm is very effective.

Lastly, in order to check the sophistication of the YOLO-DRS algorithm, we use the same network metrics to compare it to the current state-of-the-art target detection algorithms of the same class. Table 10 shows the detailed data of YOLO-DRS experiments with different advanced algorithms.

It can be seen from Table 10 that our proposed YOLO-DRS achieves the best results on the P, R, and mAP@0.5 metrics, which is sufficient to prove the speed and sophistication of our proposed algorithms.

### 5.5. Visualization Experiments

To more intuitively reflect the algorithm’s solution to the problem of detecting RSOD datasets, we performed visualization and comparison experiments in a variety of scenarios. Note that the image used for the visualization experiment is the RSOD test dataset.

Firstly, Figure 13 illustrates small-target missed detection and false detection.

Obviously, YOLO-DRS can solve the problem of small-target aircraft missed and false detection and improve the detection accuracy of the model.

Secondly, Figure 14 demonstrates the average accuracy of detection of the target. It is not hard to see that YOLO-DRS is able to achieve improved detection accuracy.

Thirdly, Figure 15 illustrates the detection results of the target on multiple scales. The background information in Figure 15 is complex and the aircraft types are characterized by a multi-scale distribution. YOLO-DRS can greatly reduce the problem of missed detection.

Finally, Figure 16 illustrates the detection of large scales with complex background information. YOLO-DRS still performs well in detecting large-scale targets and is able to reduce the problem of false detection of large-scale targets under complex background information.

In conclusion, the YOLO-DRS proposed in this paper can solve the problems of low average detection accuracy, false detection, and missed detection caused by the characteristics of many small objects in remote sensing images, with multi-scale distribution and complex background information.

## 6. Conclusions

Based on YOLOv5s, we propose YOLO-DRS, a lightweight remote sensing image object detection algorithm that fuses multiple scales efficiently. Firstly, we propose an efficient and lightweight multi-scale attention mechanism, LEC, that is able to capture multi-scale target features under complex background information with little computational overhead. Then, we introduce the transposed convolutional replacement nearest-neighbor upsampling algorithm, which can dynamically learn the feature information and can reduce the loss of target feature information during the upsampling process.

On the RSOD dataset, we obtained 97.5% mAP@0.5, an improvement of 2.5% over the original YOLOv5s, and only an increase of 0.2 in FLOPs. In addition, YOLO-DRS improves the mAP@0.5 metrics number by 1.8% and 7.3% compared to the state-of-the-art algorithms YOLOv8s and YOLOv7-tiny, respectively. In summary, the YOLO-DRS algorithm is able to solve the problem of low average accuracy of detection, false detection, and missed detection in remote sensing images due to the complex background information, many-small-target multi-scale distribution, and other characteristics. Moving forward, we will explore the study of pruning and lightweighting the model without degrading the detection accuracy so that it can be better deployed for grounded applications. 

## Figures and Tables

**Figure 1 biomimetics-08-00458-f001:**
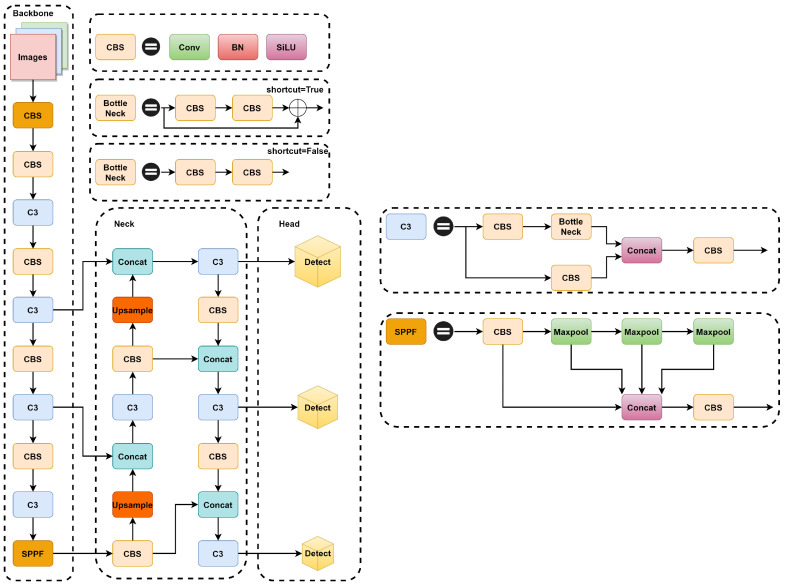
Structure of YOLOv5s.

**Figure 2 biomimetics-08-00458-f002:**
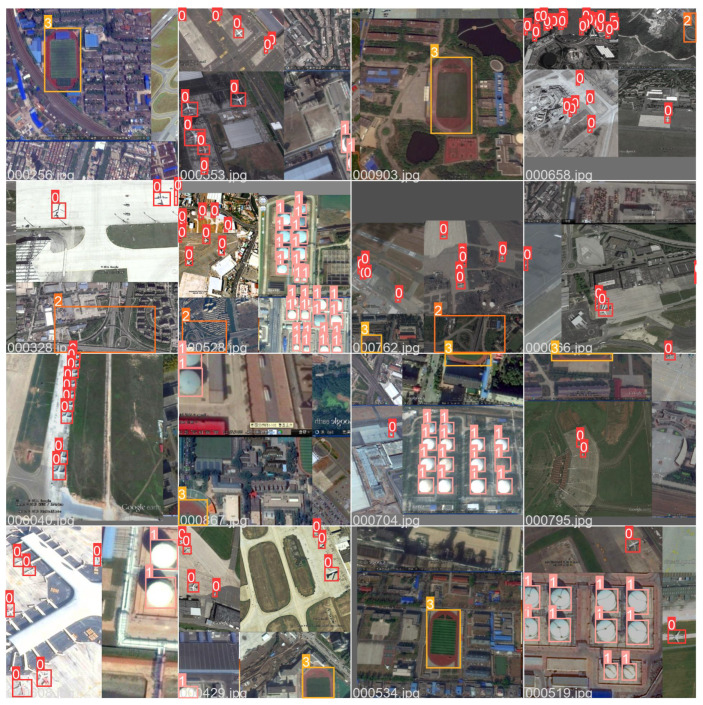
Mosaic4 Enhanced Image.

**Figure 3 biomimetics-08-00458-f003:**
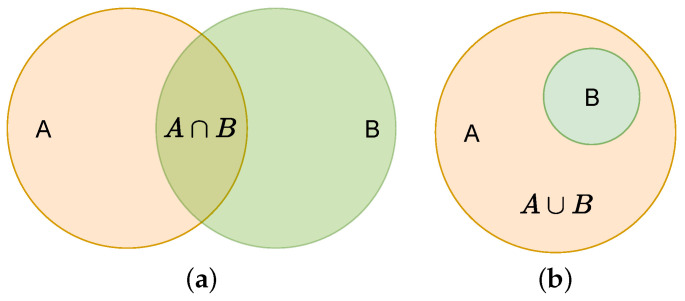
IOU loss. (**a**) A∩B is the intersection. (**b**) A∪B (Equal to A) is the union.

**Figure 4 biomimetics-08-00458-f004:**
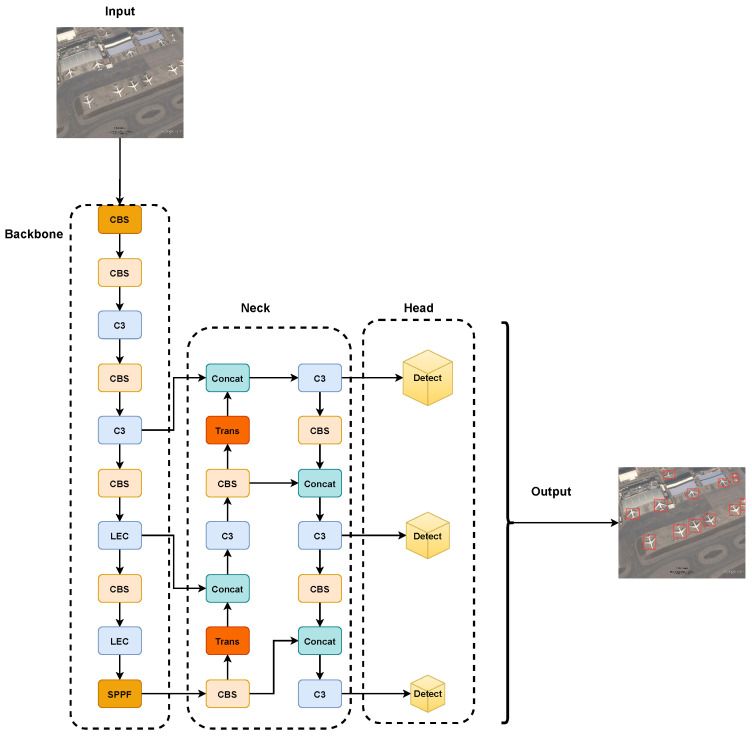
Structure of YOLO-DRS. (Trans is transposed convolution).

**Figure 5 biomimetics-08-00458-f005:**
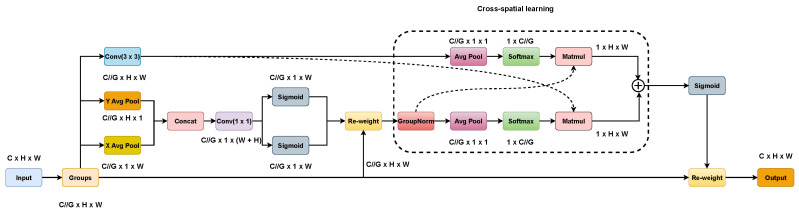
Structure of EMA.

**Figure 6 biomimetics-08-00458-f006:**
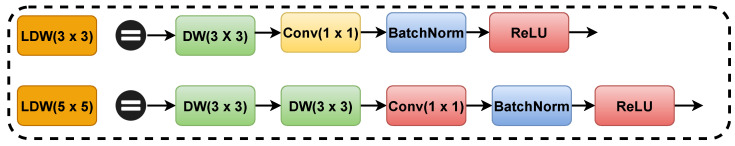
Structure of LDW convolutional block (DW is depthwise convolution).

**Figure 7 biomimetics-08-00458-f007:**
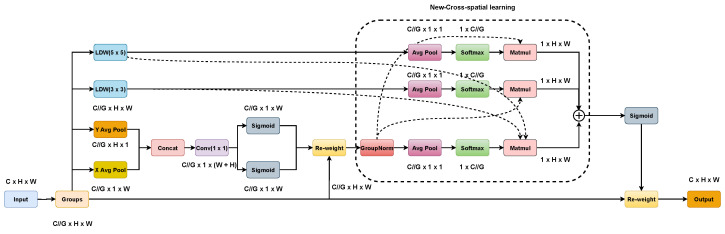
Structure of LDW-EMA.

**Figure 8 biomimetics-08-00458-f008:**
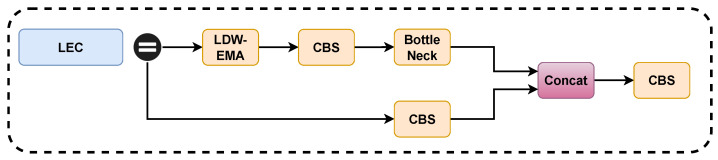
Structure of LEC Module.

**Figure 9 biomimetics-08-00458-f009:**
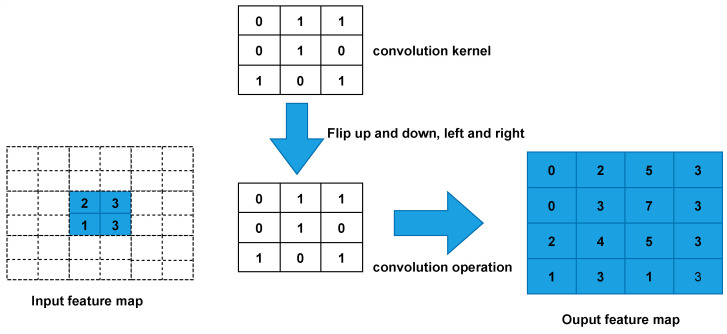
Transposed convolution Computation Process.

**Figure 10 biomimetics-08-00458-f010:**
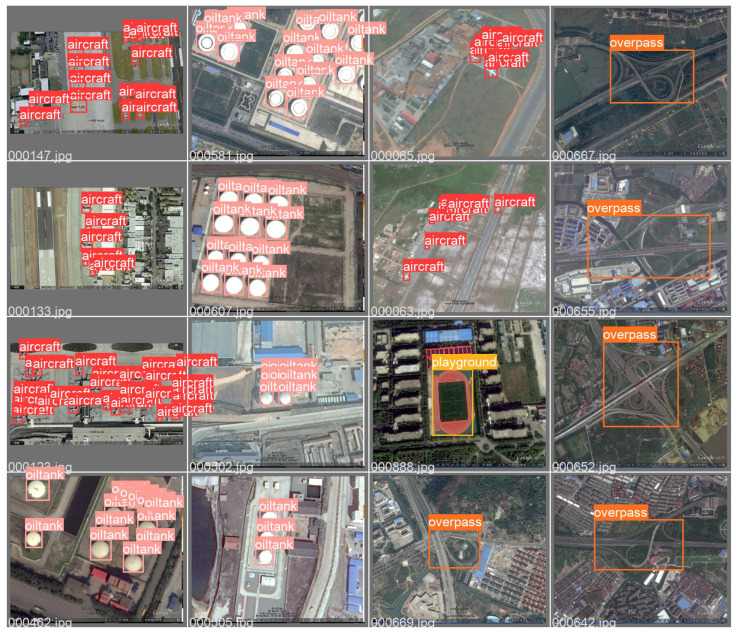
Sample Visualization of RSOD Dataset.

**Figure 11 biomimetics-08-00458-f011:**
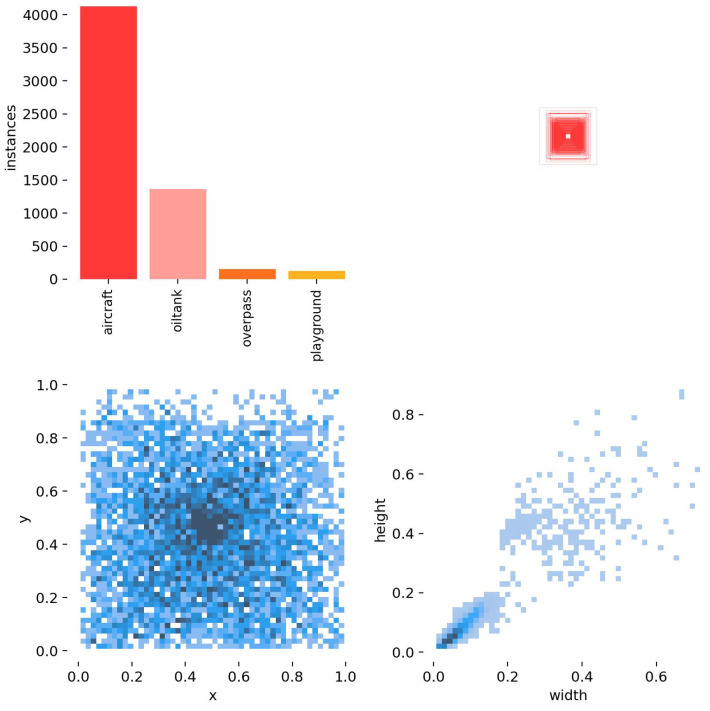
Label Information Distribution.

**Figure 12 biomimetics-08-00458-f012:**
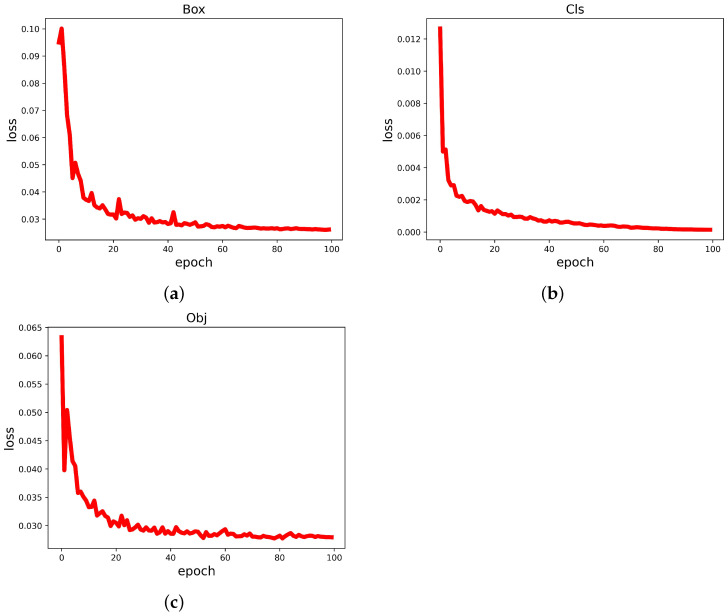
Loss Curve. (**a**) Box Loss. (**b**) Cls Loss. (**c**) Obj Loss.

**Figure 13 biomimetics-08-00458-f013:**
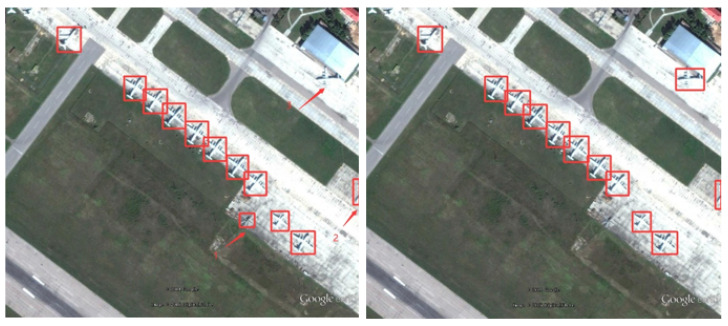
Small-target missed detection and false detection in complex backgrounds. The detection results of YOLOv5s are shown on the **left** and the results of YOLO-DRS are shown on the **right**.

**Figure 14 biomimetics-08-00458-f014:**
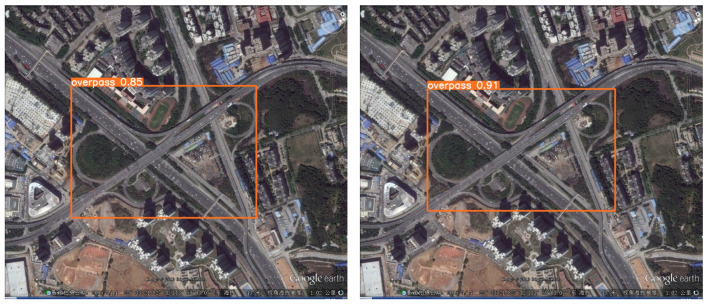
Detection accuracy of models in complex backgrounds. The detection results of YOLOv5s are shown on the **left** and the results of YOLO-DRS are shown on the **right**.

**Figure 15 biomimetics-08-00458-f015:**
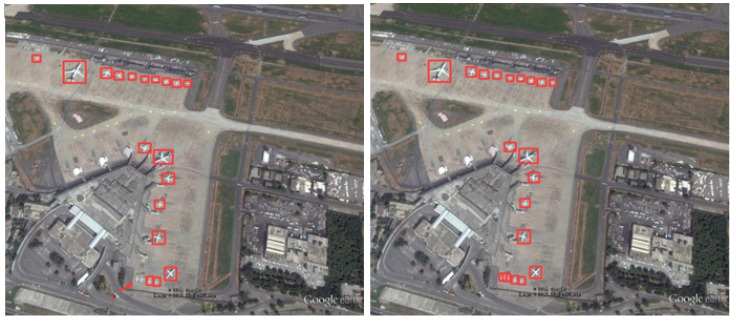
Multi-scale small-target missed detection in complex backgrounds. The detection results of YOLOv5s are shown on the **left** and the results of YOLO-DRS are shown on the **right**.

**Figure 16 biomimetics-08-00458-f016:**
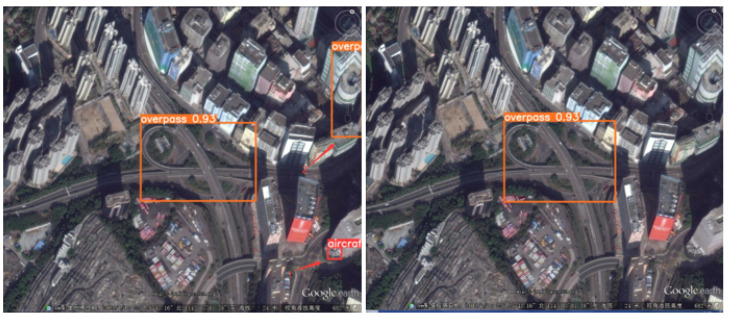
Large-target false detection in complex backgrounds. The detection results of YOLOv5s are shown on the **left** and the results of YOLO-DRS are shown on the **right**.

**Table 1 biomimetics-08-00458-t001:** Experimental Environment Configuration.

Project	Environment
Operating System	Ubuntu
CPU	E5-2680 v4
GPU	GeForce RTX 3060
Memory	14 GB
Pytorch version	1.10.0
CUDA	11.1

**Table 2 biomimetics-08-00458-t002:** Distribution of Datasets.

Dataset Labeling	Number of Images
aircraft	446
playground	189
overpass	176
oiltank	165

**Table 3 biomimetics-08-00458-t003:** Experimental Parameter Setting.

Parameters	Value
weights	yolov5s.pt
division ratio	7:2:1 (train:val:test)
optimizer	SGD
batch size	16
epochs	100

**Table 4 biomimetics-08-00458-t004:** Comparison of LEC Experiments at Different Locations.

Method	Precision	Recall	mAP@0.5	FLOPs
YOLOv5s	0.930	0.939	0.950	15.8
+LEC-top	0.959	0.956	0.972	15.9
+LEC-bottom	0.962	0.925	0.970	15.9
+LEC-both	0.967	0.936	0.963	16.0

**Table 5 biomimetics-08-00458-t005:** Experimental Comparison of Different Loss Functions.

Method	Precision	Recall	mAP@0.5	FLOPs
YOLOv5s	0.930	0.939	0.950	15.8
+LDW (ReLU)	0.959	0.956	0.972	15.9
+LDW (SiLU)	0.953	0.975	0.967	15.9
+LDW (Mish)	0.961	0.952	0.968	15.9

**Table 6 biomimetics-08-00458-t006:** Experimental Comparison of Transposed Convolution at Different Positions.

Method	Precision	Recall	mAP@0.5	FLOPs
YOLOv5s	0.930	0.939	0.950	15.8
+Trans-first	0.963	0.939	0.961	15.8
+Trans-second	0.966	0.936	0.975	15.8
+Trans-both	0.959	0.971	0.971	15.8

**Table 7 biomimetics-08-00458-t007:** Experimental Comparison of Different EMA Attention Mechanisms.

Method	Precision	Recall	mAP@0.5	FLOPs
YOLOv5s	0.930	0.939	0.950	15.8
+EMA	0.937	0.957	0.956	15.8
+LDW (3 × 3)	0.921	0.960	0.964	15.8
+LDW-EMA	0.959	0.956	0.972	16.0

**Table 8 biomimetics-08-00458-t008:** Experimental Comparison of Mainstream Attention Mechanisms.

Method	Precision	Recall	mAP@0.5	FLOPs
YOLOv5s	0.930	0.939	0.950	15.8
+C3CA	0.970	0.935	0.957	15.8
+C3ECA	0.950	0.940	0.955	15.8
+C3SE	0.990	0.923	0.968	15.8
+LEC	0.959	0.956	0.972	16.0

**Table 9 biomimetics-08-00458-t009:** Ablation Experiment.

Method	Precision	Recall	mAP@0.5	FLOPs	FPS/(frame/s)
YOLOv5s	0.930	0.939	0.950	15.8	76.8
+LEC	0.959	0.956	0.972	16.0	51.9
+Trans-both	0.959	0.971	0.971	15.8	69.7
YOLO-DRS	0.953	0.971	0.975	16.0	53.9

**Table 10 biomimetics-08-00458-t010:** Experimental Comparison of Mainstream Algorithms.

Method	Precision	Recall	mAP@0.5
YOLOv5s	0.930	0.939	0.950
YOLOv7-tiny	0.953	0.957	0.957
YOLOv8s	0.871	0.864	0.902
YOLO-DRS	0.953	0.971	0.975

## Data Availability

Not applicable.

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
