# Peer review of "YOLO-DRS: A Bioinspired Object Detection Algorithm for Remote Sensing Images Incorporating a Multi-Scale Efficient Lightweight Attention Mechanism"

_biomimetics, 2023, doi:10.3390/biomimetics8060458_

Round 1

Reviewer 1 Report

Is a period needed at the end of the paper title?

Please discuss why this paper is important for this journal.

Fig 3 B subtitle and figure contain errors. The symbol is intersection, not union.

Please explain more technical details on limitation of this work and future work.

Clarify originality and advantages against state of art.

Please discuss more relevant recent work, such as recent work on detection:

Cyber-threat detection system using a hybrid approach of transfer learning and multi-model image representation

Sensors 22 (15), 5883

Explain the generalisability of results in good technical details.

Reviewer 2 Report

The authors propose a transposed convolution up-sampling alternative to the original nearest neighbor interpolation 9 algorithm The approach is well explained and the example allows readers to understand the whole problem, and also to replicate it. Despite the paper is well presented and structured, I suggest introducing the main results in the abstract in brief, so that it is possible to have an idea about the overall ones. Moreover, the introduction, in the presented form, is quite negligible. I suggest restructuring it explaining in-depth the maintenance problem. The English form has to be revised for the presence of some typos. The literature review appears not completely updated.

good
